# Adaptogens on Depression-Related Outcomes: A Systematic Integrative Review and Rationale of Synergism with Physical Activity

**DOI:** 10.3390/ijerph20075298

**Published:** 2023-03-28

**Authors:** Isabel A. Sánchez, Jaime A. Cuchimba, María C. Pineda, Yenny P. Argüello, Jana Kočí, Richard B. Kreider, Jorge L. Petro, Diego A. Bonilla

**Affiliations:** 1Grupo de Investigación Ciencias Aplicadas al Ejercicio, Deporte y Salud—GICAEDS, Universidad Santo Tomás, Bogotá 205070, Colombia; isabel.sanchez@usta.edu.co (I.A.S.);; 2Grupo de Investigación Cuerpo, Sujeto y Educación—CSE, Universidad Santo Tomás, Bogotá 205070, Colombia; 3Research Division, Dynamical Business & Science Society—DBSS International SAS, Bogotá 110311, Colombia; janakoptikova@gmail.com (J.K.); jlpetro@dbss.pro (J.L.P.); 4Department of Education, Faculty of Education, Charles University, 11636 Prague, Czech Republic; 5Exercise & Sport Nutrition Laboratory, Human Clinical Research Facility, Texas A&M University, College Station, TX 77843, USA; rbkreider@tamu.edu; 6Research Group in Physical Activity, Sports and Health Sciences (GICAFS), Universidad de Córdoba, Montería 230002, Colombia; 7Research Group in Biochemistry and Molecular Biology, Universidad Distrital Francisco José de Caldas, Bogotá 110311, Colombia

**Keywords:** adaptogens, herbal supplements, allostasis, depression, health sciences, quality of life

## Abstract

Depression is considered the most important disorder affecting mental health. The aim of this systematic integrative review was: (i) to describe the effects of supplementation with adaptogens on variables related to depression in adults; and (ii) to discuss the potential combination with physical exercise to aid planning and commissioning future clinical research. An integrative review was developed complementing the Preferred Reporting Items for Systematic reviews and Meta-Analyses statement (PROSPERO registration: CRD42021249682). A total of 41 articles met the inclusion criteria. With a Price index of 46.4%, we found that: (i) *Hypericum perforatum* (St. John’s Wort) is the most studied and supported adaptogen (17/41 [41.46%], three systematic reviews with meta-analysis) followed by *Crocus sativus* L. or saffron (6/41 [14.63%], three systematic reviews with meta-analysis and two systematic reviews); (ii) it is possible that the significantly better performance of adaptogens over placebo is due to the reduction of allostatic load via the action of secondary metabolites on BDNF regulation; and, (iii) the number of studies reporting physical activity levels is limited or null for those that combine an exercise program with the consumption of adaptogens. Aware of the need for a multidisciplinary approach for depression treatment, this systematic integrative review provides an up-to-date view for supporting the use of St. John’s Wort and saffron as non-pharmacological strategies while also help commissioning future research on the efficacy of other adaptogens. It also contributes to the design of future clinical research studies that evaluate the consumption of herbal extracts plus physical exercise, mainly resistance training, as a potentially safe and powerful strategy to treat depression.

## 1. Introduction

Depression is considered the most important disorder affecting mental health, being associated with an increased risk of premature mortality and other diseases [1]. Depression is defined as a state where the person only holds negative perceptions of the environment and him/herself, experiencing a deterioration in their emotional state until significant impacts are felt in their daily activities at work and personal life [2]. There are several risk factors, including stress, frustration, and a low level of physical activity; in this regard, a high prevalence rate of depression (up to 48.3%) has been reported worldwide during SARS-CoV-2 confinement [3].

Depression is associated with dysfunction of different neurotransmitters, such as serotonin, noradrenaline, and dopamine [4,5,6]. At the tissue level, the existence of alterations in brain regions such as frontal lobes and the hypothalamus has been evidenced [7]. Considering these neurophysiological changes during the development of depression, it is important to evaluate the biological mechanisms of adaptation to stressors. The most adequate model at present for this is known as allostasis, considered a predictive regulation model [8]. Upon exposure to a stressor, any biological system integrates different complex mechanisms that use prior knowledge, make predictions, and acquire further new knowledge to adapt before the next eventualities. In the case of the nervous system, these mechanisms include (i) steroid hormones and peptides that regulate the physiological response; (ii) neural signals that generate cross-talk between different areas of the brain; and (iii) brain stem nuclei raphe signals that modulate levels of arousal and mood through a main neurotransmitter, serotonin [9]. It is important to emphasize that repeated exposure to a stressor leads to the so-called allostatic load, which must be necessary to facilitate adaptation and prevent the disruption of systemic regulators (thresholds depend on each biological system) [10,11]. The allostatic load is defined as the additional energetic burden that the organism must bear to adapt and survive [12]. In this sense, the allostatic model allows us to analyze and discuss changes evidenced during the development of depression, thus implementing the monitoring and diagnosis of the disease at different timescales as described recently [13]. Interestingly, Barrett, et al. [14] proposed that the core of depression is a disorder of metabolism and energy regulation (i.e., allostasis overload) with sensory consequences of that regulation (i.e., altered interoception) which results in a relatively ‘locked-in’ depressed brain. This seems to be a valid integration under the paradigm of interoception as modelling and allostasis as control [15]. In fact, previous studies have shown that the greater the allostatic load, the more severe the depressive symptoms in older adults [16].

To deal with depression, several non-pharmacological treatments have been studied including cognitive behavioral therapy, naturopathic interventions, psychotherapy, and physical exercise-based interventions [17]. Under the unified theory, these strategies address the three potential ingredients that can contribute to depression: (i) a metabolically inefficient internal model; (ii) unreliable prediction errors; and, (iii) inaccurate precision signals [14]. Interestingly, due to the positive effects of increasing the level of physical activity on different components of quality of life (e.g., physiological, psychological, and social) [18,19], more and more studies are supporting physical exercise as an evidence-based treatment for depression [20,21,22,23]. In fact, physical activity is considered a mechanism to reduce the allostatic load in several populations [24].

Although less studied, traditional Chinese medicine and Ayurveda have long been used as complementary therapies for the treatment of various diseases, including depression. Among the most common practices is the use of herbal extracts and adaptogens [25]. The term adaptogen was first proposed in 1940 by the Soviet toxicologist Lazarev to describe some herbs that can non-specifically enhance the human body [26]. Subsequently, Brekhman and Dardimov [27] proposed in 1964 a broader version by referring to adaptogens as harmless agents, which non-specifically increase resistance to harmful factors (“stressors”) of a physical, chemical, biological and psychological nature. These pharmacologically active compounds have been shown to elicit a state of non-specific resistance that allows the organism to counteract stressors and/or promote physiological adaptation [28]. Currently, there are a large number of plant-based adaptogens with potential properties for the treatment of neuropathologies, including depression: *Panax* (ginseng), *Hypericum perforatum*, *Pfaffia paniculata*, *Rauwolfia serpentina*, *Rhodiola Rosea*, *Withania somnifera* (Ashwagandha), *Eleutherococcus senticosus*, *Centella asiatica*, *Camellia sinensis*, *Astragalus*, *Valeriana officinalis*, *Schisandra chinensis*, *Lepidium meyenii* (Maca) and *Cordyceps* [29]. Most of the plant extracts are characterized by the presence of secondary metabolites with antioxidant and anti-inflammatory effects. These compounds are synthesized by the plants and comprise four major groups such as terpenoids, alkaloids, glycosinolates and phenols [30]. Despite the above, further research is required to establish the most effective plant-based adaptogens, their doses and recommended protocols as co-adjuvants for the treatment of depression. Likewise, the evaluation of the potential combination of adaptogens and physical activity as a non-pharmacological treatment of depression warrants research. Thus, the aim of this systematic integrative review was (i) to evaluate the effect of adaptogen supplementation on depression-related symptoms and outcomes in adults (>18 years old), and (i) to discuss the potential combination with physical exercise to aid planning and commissioning future clinical research.

## 2. Materials and Methods

The literature review followed the basic framework for integrative reviews described by Whittemore and Knafl [31], which allowed the inclusion of quantitative and qualitative studies. In addition, we used the optimized methodology established by Hopia et al. [32] for the evaluation and analysis of scientific publications, including problem formulation, a literature search, evaluation, analysis, and presentation of findings. The systematic integrative approach has been used previously in health-related topics [33,34,35]. The results were reported according to the established guidelines of the preferred reporting items for systematic reviews and meta-analyses (PRISMA) guidelines [36].

### 2.1. Eligibility Criteria

The inclusion criteria for this systematic review were as follows: (1) clinical trials (randomized or not), case studies, and comprehensive systematic reviews; (2) published from 2010 onwards; (3) written in English or Spanish; (4) full text available; and (5) studies that evaluated the effect on depression-related symptoms and outcomes (i.e., rating scales, biomarkers and/or diagnostic imaging techniques) after supplementation or co-supplementation with *Panax* (ginseng), *Hypericum perforatum*, *Pfaffia paniculata*, *Rauwolfia serpentina*, *Rhodiola Rosea*, *Withania somnifera* (Ashwagandha), *Eleutherococcus senticosus*, *Centella asiatica*, *Crocus sativus* (Saffron), *Camellia sinensis*, *Astragalus*, *Valeriana officinalis*, *Schisandra chinensis*, *Lepidium meyenii* (Macca) and *Cordyceps*.

### 2.2. Information Sources

The following academic research databases were selected to explore the literature: PubMed/MEDLINE, ScienceDirect, Scopus, Cochrane, EMBASE, SciELO, OVID and Google Scholar. Further papers were sought manually.

### 2.3. Search Strategy

The patient, intervention, comparison, outcome (PICO) model was utilized for structuring our research question: P (subjects aged > 18 years old with symptoms and/or diagnosed depression) I (supplementation with adaptogens) C (placebo, or non-exposed control group [pre-post]) O (depression-related symptoms and outcomes) [37]; however, we used the truncated PIC approach to build the search algorithm to emphasize the comparison intervention or exposure [38]. Three authors (M.C.P., J.A.C., and D.A.B.) performed the search independently using the following Boolean algorithms: PubMed, Cochrane Library, Scielo, *(adaptogen* OR panax OR ginseng OR Hypericum perforatum OR St John’s wort OR pfaffia paniculata OR rauwolfia serpentina OR rhodiola OR withania somnifera OR ashwagandha OR eleutherococcus OR centella asiatica OR camellia sinensis OR astragalus OR valeriana officinalis OR schisandra chinensis OR lepidium meyenii OR maca OR cordyceps) AND depress**; ScienceDirect, EMBASE and Ovid, *(adaptogen OR panax OR ginseng OR Hypericum perforatum OR St John’s wort OR pfaffia paniculata OR rauwolfia serpentina OR rhodiola OR withania somnifera OR ashwagandha OR eleutherococcus OR centella asiatica OR camellia sinensis OR astragalus OR valeriana officinalis OR schisandra chinensis OR lepidium meyenii OR maca OR cordyceps) AND depress*. Additionally, the identification of potential studies was enriched by performing a manual search in Google Scholar with free language terms related to “adaptogens, herbal extracts and depression”.

### 2.4. Selection Process

The selection was carried out by four researchers (M.C.P., J.A.C., I.A.S. and D.A.B.) under the supervision of the other co-authors. After the search of published articles, the filter options of the databases were used to meet inclusion criteria 1 to 4. After this searching process, the remaining references were manually filtered in an Excel file by screening the title and abstract in order to identify duplicates and those ineligible after evaluation of criterion 5. The selection process took place during October 2021 and July 2022, although an updated search was conducted prior to manuscript submission.

### 2.5. Data Collection Process and Items

The full-text articles of the selected studies were evaluated for meeting the inclusion criteria. The following data were obtained and analyzed from the selected quantitative studies: (i) descriptive statistics of the study population; (ii) study length; (iii) characteristics of the adaptogen supplementation protocol; (iv) the analyzed variables; (v) significant differences in comparison to placebo or non-exposed control group (where available); and (vi) study conclusions. In the case of qualitative studies, we extracted the objective, the type of study, the methodology, and the conclusions by the authors.

### 2.6. Study Risk of Bias Assessment

Two authors independently evaluated the risk of bias of all included clinical trials using the Cochrane risk of bias tool RoB 2.0 [39]: selection bias, performance bias, detection bias, attrition bias, reporting bias, and any other bias. All randomized participants in the analysis were included, as it was the least biased way to analyze clinical effects. Discrepancies were identified and resolved through discussion (with a third author where necessary). The figures to summarize the results of the risk of bias assessment were developed using the Risk-of-bias VISualization tool (robvis) [40].

### 2.7. Data Synthesis

Administered and self-reported questionnaires for depression allow evaluation of the symptomatological profile and to determine the severity of the depressive picture. The two most relevant hetero-applied questionnaires were (i) the Hamilton Rating scale for Depression (HAMD), which is of great practical value for evaluating the results of treatment, and is one of the most widely used [41]; and (ii) the Montgomery–Asberg scale, which has an advantage in excluding anxiety symptoms [42]; for this reason, several studies have demonstrated its validity and have classified it as the most important screening for the diagnosis of depression [43]. On the other hand, self-administered scales allow for easier data collection, including but not limited to: (i) the Beck Depression Inventory (BDI) [44]; (ii) the inventory of state-trait depression (IDER) that evaluates the positive (euthymia) and negative (dysthymia) affective state; (iii) the Geriatric Depression Scale (GDS) as a valid and reliable tool for the diagnosis of depression in older adults [45], (iv) the Teate Depression Inventory (TDI-E), which has been shown to exceed the accuracy of HAMD and BDI by 50% [46]; and (v) the Zung self-assessment scale [47].

Although we synthesized effects based on these rating scales, we also considered biomarkers (e.g., brain-derived neurotrophic factor [BDNF], tumor necrosis factor α [TNF-α], interleukin-6 [IL-6], serotonin levels and receptors [5-HT], etc.) and image diagnosis (e.g., magnetic resonance imaging [RMN], computerized axial tomography [CAT], positron emission tomography [PET]) if available. We extracted the changes from the baseline (mean change and SDs of the changes) in the experimental interventions (Adaptogens) and the comparator interventions (placebo or non-exposed control). In addition, the physical activity level of the studied population was set as covariable and extracted from the article information if available. A table of results and comparisons of findings was developed and complemented by the review authors considering the items mentioned before.

## 3. Results

### 3.1. Study Selection

The initial search with Boolean algorithms retrieved a total of 4911 articles. However, after filtering the publications (duplicates and inclusion criteria assessment), only 47 were potentially eligible articles. A total of 41 articles met the inclusion criteria (Figure 1).

### 3.2. Risk of Bias within Studies

The methodological quality of the selected studies included in this systematic integrative review is presented in Figure 2. In general, a moderate risk of bias was detected among the included articles.

### 3.3. Results of Individual Studies

Table 1 presents the main results of the included studies.

## 4. Discussion

### 4.1. Summary of Evidence

Pharmacological and clinical studies evaluating adaptogenic herbal extracts or secondary metabolite-enriched mixtures have shown a positive impact on human physiology under stressful situations. We aimed to perform a systematic integrative review on the effects of several adaptogens on depression-related outcomes besides evaluating the potential combination with physical exercise to optimize results. Similar to previous reports in peri- and postmenopausal women [89] and healthy individuals [90], it has been found that certain adaptogens might have low-to-moderate clinical effects to relieve anxiety and mild depression [31,32,33,34,35,36,37,38,39,40,41,42,43,44,45,46,47,48,49,50,51,52,53,54]. It is important to note that the number of articles reporting the level of physical activity is very low and null for those combining an exercise program with adaptogen consumption.

A total of 5922 predominately female adults (aged 18 to 97) participated in the clinical trials analyzed in this integrative review (a Price index of 46.4% was obtained for the selected studies). Most of the participants were outpatients diagnosed with mild to moderate major depressive disorder. Importantly, the *Hypericum perforatum* extract was the most studied adaptogen in the sample of included articles (17/41 [41.46%], three systematic reviews with meta-analysis), followed by *Crocus sativus* L. (6/41 [14.63%], three systematic reviews with meta-analysis and two systematic reviews), *Rhodiola rosea* (6/41 [14.63%], two systematic reviews), *Withania somnifera* (Ashwagandha, 5/41 [12.19%]), *Panax* (as Korean red ginseng, 3/41 [7.31%]), *Lepidium meyenii* (Macca, 1/41 [2.43%]), *Valeriana officinalis* (1/41 [2.43%]), and *Cordyceps* (1/41 [2.43%]). It is worth noting that previous meta-analyses have been performed before the inclusion criterion for the date of publication of this systematic integrative review. For example, a meta-analysis of clinical trials performed by Linde and colleagues in 2008 indicated *Hypericum perforatum* extract to be superior to a placebo in patients with major depression, to be similarly effective, and to have fewer side effects than standard antidepressants [91]. Thus, the volume of clinical research and meta-analytic data [69,71,88] reinforces the fact that *Hypericum perforatum* is the most studied, probably safe and effective adaptogenic herbal extract that might be recommended to practitioners as part of an integral clinical treatment to reduce depressive symptoms. Nevertheless, more research is warranted to evaluate factors that influence response to treatment, such as the type of extract, the duration of the disorder and age of the participants, among others. *Crocus sativus* L. also has a substantial body of meta-analytic evidence to support clinical practice [57,72,79]; unfortunately, saffron is one of the most expensive herbs on the market today, which reduces its accessibility and practical application [92]. The meta-analyses that evaluated the effects of *Hypericum perforatum* and *Crocus sativus* L. were performed in different research groups from several countries including the USA, Singapore, China, Australia, and Hungary.

Contrary to common beliefs, the clinical trials included in this study have been carried out in non-Asian countries such as the USA (7), Germany (5), the UK (3), Canada (1), and Australia (1). This might be seen as an effort to corroborate findings of the last decades by research groups from Korea, China, Iran, and India. In fact, re-analyses of published data are also frequently found in the included studies with the work by Grobler et al. [58] as the only re-analysis that highlighted contradictory findings to the original paper. Large and multi-centered scientific studies have mainly been performed in Germany and the UK. There was a high prevalence of depression assessment using rating scales (e.g., HAMD, BDI) with no systematic reports of biomarkers or imaging analysis in the selected studies. Overall, clinical trials on herbal adaptogenic extracts showed significant improvement in depression-related outcomes over placebo or baseline values (Table 1)—except for *Cordyceps militaris*. No articles that evaluated the effects of *Pfaffia paniculata*, *Rauwolfia serpentina*, *Eleutherococcus senticosus*, *Centella asiatica*, *Camellia sinensis*, *Astragalus*, or *Schisandra chinensis* met the inclusion criteria. Interestingly, the findings of the cumulative evidence suggest other additional effects such as the restoration and improvement of mental energy, compensation for the effects of sleep deprivation, protection of the nervous system, and the improvement of memory and perception [52,84,87].

### 4.2. Potential Mechanisms

Reported clinical improvements of adaptogens on depressive symptoms are probably due to the positive impact of secondary metabolites (e.g., terpenoids, alkaloids, glycosinolates, and phenols) on cellular allostasis (Figure 3). Todovora et al. (2022) [93] concluded that phytostanols, phytosterols, alkaloids, and saponins are among the main phytochemicals isolated from several adaptogenic herbal extracts. Indeed, it is proposed nowadays that depression arises from chronic energy inefficiency and an altered default mode network of the nervous system (interoceptive signaling) [13,14]. In many cases, the antioxidant and anti-inflammatory properties against oxidative stress of the adaptogen-derived secondary metabolites (or other stressors common in depression) have been highlighted to play a protective role at the neurocellular level [30,93,94,95]; however, further research is needed to clarify the metabolic pathways in each adaptogenic herbal extract. In the next paragraphs, we elaborate a brief description of the potential mechanisms of action of the most studied adaptogens (*Hypericum perforatum*, *Crocus sativus*, *Rhodiola rosea*, *Panax*, and *Withania somnifera*) and the metabolic signatures of depression.

According to Marrelli et al. (2020), *Hypericum* spp. generate several secondary metabolites such as phenolic acids, proanthocyanidins, flavonoids, naphthodianthrones, acylphloroglucinols, xanthones, and essential oils [96]. Specifically, *Hypericum perforatum* has been reported to contain naphthodianthrones, phloroglucinols, xanthones, and flavonoids [97]. Likewise, the secondary metabolites of *Withania somnifera* (Ashwagandha) (i.e., 12 alkaloids and 35 withanolides) are possibly responsible for the reduction in depression symptoms [98]. Two major active components of *Hypericum perforatum*, hyperforin and hypericin, are proposed to induce a dualistic modulation of the activity of cholinergic signaling, which can be an interesting topic for future studies [99]. Studies using several models have shown that secondary metabolites derived from *Hypericum perforatum* possibly increase BDNF or activate its signaling pathway [100,101,102,103]. *Crocus sativus* L. (Saffron), as one of the most clinically effective herbal extracts [57,72,79], has shown antidepressant effects in various experimental depression models by modulating the BDNF, cyclic AMP response element binding protein (CREB), and VGF pathways [104].

Similar to other adaptogens, it has been proposed that *Rhodiola rosea* might alleviate depression via the BDNF/TrkB-GSK-3β signaling pathway [105]. Indeed, a systematic review with meta-analysis of pre-clinical studies concluded that possible mechanisms of action of *Rhodiola rosea* on ischemic stroke may be mainly due to the activation of anti-inflammatory, anti-apoptosis, and anti-oxidative pathways [106]. Complementary, saponins from *Rhodiola rosea* have been also described as important molecules since they may have potential sedative and hypnotic effects by modulating serotonergic, GABAAergic, and immune systems [107]. In the case of *Panax* spp., a recent meta-analysis of pre-clinical research reported that many ginsenosides, including ginsenoside Rg1, might be responsible for the antidepressant effects by modulating different biological mechanisms in a dose-response manner [108]. Ginsenoside Rb1, another active compound found in ginseng and water extract of *Panax ginseng*, has been shown to exert promising antidepressant-like effects in a depression model via the BDNF/TrkB-GSK-3β signaling pathway [109,110]. In addition, recent findings from in vivo studies have shown that ginsenoside Rg1 might protect against neuroinflammation by means of the suppression of connexin43 ubiquitination [111,112]. Connexin43 has an important role in the gap junction channels between astrocytes as the most abundant connexin expressed in these cells; however, a higher ubiquitin-mediated degradation of connexin43 has been associated with the concomitant progression of depression [113]. Interestingly, Korean red ginseng has been proposed to alleviate depressive disorder by improving astrocyte gap junction intercellular communication [114] and which is linked to reductions in neuroinflammation [115]. Nevertheless, this novel anti-depressant mechanism remains fully unknown and warrants more research.

Thus, as depicted in Figure 3, hippocampal BDNF signaling seems to be a conserved antidepressant mechanism among the effects of different plant species [116,117,118]. In addition, based on the analysis of genome-wide effects in T98G neuroglia cells performed recently by Panossian et al. (2017) [94], it seems that the adaptogenic herbal extracts—including those reviewed in this article—might play a modulating role on adaptive stress–response signaling pathways (ASRSPs) such as (i) corticotropin-releasing hormone, cAMP-mediated protein kinase A, and CREB; (ii) signaling pathways related to CXCR4, melatonin, nitric oxide synthase, GP6, Gαs, MAPK, neuroinflammation, neuropathic pain, opioids, renin-angiotensin, AMP-activated protein kinase (AMPK), calcium, and synapses; (iii) and biological processes such as dendritic cell maturation and G-coupled protein receptor-mediated nutrient sensing in enteroendocrine cells.

### 4.3. Rationale for Synergism with Physical Activity and Future Directions

Future studies integrating adaptogen consumption and a physical exercise program are expected. Regular practice of physical activity has demonstrated a high impact on both physical and mental health in humans [119,120]. These effects point to physical exercise as a protective and inducing agent of biochemical agents responsible for aspects such as the speed of information processing, memory, attention, neuronal signaling, and coordination [121]. In relation to this, Matias et al. [122] reported that physical activity is associated with reduced prevalence of diagnosed depressive symptoms. Currently, physical exercise (both cardiovascular and strength exercise training) is seen as an effective treatment that is associated with significant reductions in and less frequency of symptoms in patients with clinical management for depression [20,21,22].

It must be noted that some nutrients have also been linked or used as potential non-pharmacological strategies during the integral treatment of depression. Kamalzadeh et al. [123] and Okereke [124] have reported an important inverse association between dietary intake of vitamin D and the reduction of depressive symptoms. These anti-depressant effects are enhanced by supplementation with vitamin D3, especially if combined with physical exercise [125]. Similarly, Kious et al. [126] and Ostojic et al. [127] have highlighted the use of creatine as a powerful element for the production and storage of brain energy that can be considered in the form of creatine monohydrate as a safe nutritional strategy for the treatment of depression. It is necessary to point out that depression prevalence is 42% higher among U.S. adults that consume less dietary creatine (0–0.26 g/day) in comparison to those with higher creatine intake (0.70–3.16 g/day) [128]. This agrees with and reinforces the current hypothesis of depression etiology that gives relevance to the dysregulation of energy production in the brain (as part of the allostatic overload) [14].

Dosed physical activity might be considered an effective strategy to reduce allostatic load [24], even in depression and anxiety disorders [129]. Adaptogens have been also associated with lower allostatic load in stressful conditions [130,131]. Interestingly, from a mechanistic point of view, both adaptogenic herbal extracts and physical exercise share the heat shock proteins (HSP) as characteristic modulators of their cellular physiological response–adaptation processes [132,133]. The role of exercise in depression has been also linked to the BDNF pathway and a variety of immuno-inflammatory mechanisms [134] similar to those specifically described for the adaptogenic herbal extracts in relation to the stress-protective activity and increased adaptability of the organism [95]. Considering this, we hypothesize possible additive effects of combining exercise and certain adaptogenic herbal extracts. We suggest that the consumption of Hypericum perforatum or Rhodiola rosea during an exercise physical training program (either resistance or endurance) might evoke positive health outcomes and a significant decrease in depression scores. In spite of the meta-analytic evidence supporting its use, we did not include *Crocus sativus* L. (saffron) in this practical recommendation due to its high cost which reduces accessibility [135]. Furthermore, it is worth noting the potential impact of *Withania somnifera* (Ashwagandha) supplementation since it has been recently concluded in systematic reviews with meta-analysis to be an effective strategy to improve physical function [136] and to relieve stress and anxiety [137] in different populations.

### 4.4. Limitations

This systematic integrative review should be read in light of the following limitations. Firstly, the analysis of the effectiveness of treatments for major psychiatric disorders (including depression) is a highly complex and responsible subject, which requires an interdisciplinary team composed of psychiatrists, pharmacologists, and epidemiologists, among others. In this regard, possible treatments for major depressive disorder (a huge challenge in public health) need very clear scientific support based on solid clinical research [138]. Thus, a critical limitation of the revised literature is that each herbal extract requires extensive research and multidisciplinary insights on clinical guidance/applications, since none of these adaptogens will cure depression, which is multifactorial in nature. Secondly, even though the Jadad scale has been popular and often chosen to evaluate reporting quality, we adhere to previous recommendations and encourage researchers to use the Cochrane risk of bias tool RoB 2.0 when performing systematic reviews and meta-analyses [139]. This is noted in the systematic reviews carried out within recent years. Thirdly, although adaptogenic herbal extracts may modulate ASRSPs, special attention should be paid to the potential herb–drug interactions [140]. Other extracts from Chinese herbal medicine [141,142] with potential anti-depressant effects were not covered in this study and warrant further investigation. Finally, the standardization of herbal extract concentrations is a current problem that highly influences clinical results [143].

On the other hand, it is noteworthy that measurement scales for depression can be hetero- or self-administered, the difference being whether they are administered by a health professional. In terms of biomarkers, several molecules associated with depression are currently used as part of the diagnosis. Several studies relate a decrease in BDNF levels to the presence of depressive symptoms and disorders [144]. Other common biomarkers are certain growth factors (e.g., vascular endothelial growth factor [VEGF], fibroblast growth factor 2 [FGF2], insulin-like growth factor 1 [IGF1]) and pro-inflammatory substances (e.g., IL-6, TNF-α) [145]. Finally, diagnostic imaging encompasses the exploration of changes in certain areas of the brain (e.g., hippocampus and prefrontal cortex) in people with depression using functional RMN, diffusion tensor imaging, CAT, PET, functional near-infrared spectroscopy, among others [46,146,147,148]. Thus, we encourage researchers to commission future studies under an integral assessment approach that includes different methodologies for the diagnosis/description of depressive symptoms including biomarkers and diagnostic imaging techniques.

## 5. Conclusions

The findings of this systematic integrative review provide more evidence for continuing research on the consumption of *Hypericum perforatum* and *Crocus sativus* L. extracts as potential non-pharmacological strategies to alleviate depressive symptoms in the adult population (>18 years) as part of an integral treatment. The potential of *Rhodiola rosea*, *Panax* (Ginseng), *Lepidium meyenii* (Macca), and *Withania somnifera* (Ashwagandha) should be noted, but more evidence from well-controlled and large-scale clinical trials is crucial to establishing recommendations for these adaptogens into common practice. Future research should also focus on high-quality standardized preparations to provide pharmacovigilance data. On the other hand, considering the lack of studies evaluating the combination of adaptogens and physical exercise, the data presented in this review contribute to the design of controlled, randomized, double-blind clinical trials of the consumption of adaptogenic herbal extracts accompanied by a physical exercise program, mainly resistance training, as a potential treatment of depression.

## Figures and Tables

**Figure 1 ijerph-20-05298-f001:**
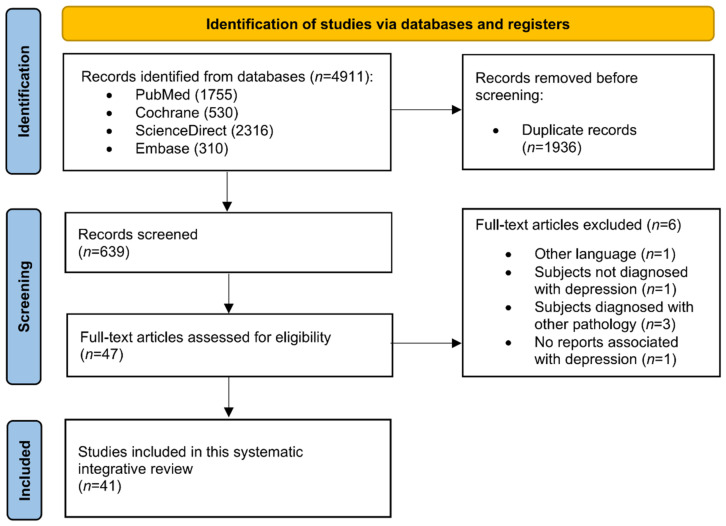
PRISMA flow Diagram.

**Figure 2 ijerph-20-05298-f002:**
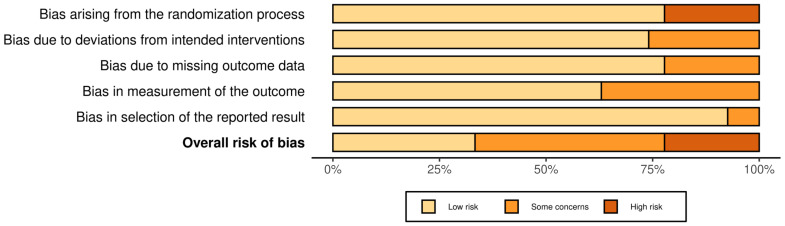
Risk of bias summary for included studies. Bar chart of the distribution of risk-of-bias judgments. The plot was obtained using the ‘robvis’ package within the R statistical computing environment.

**Figure 3 ijerph-20-05298-f003:**
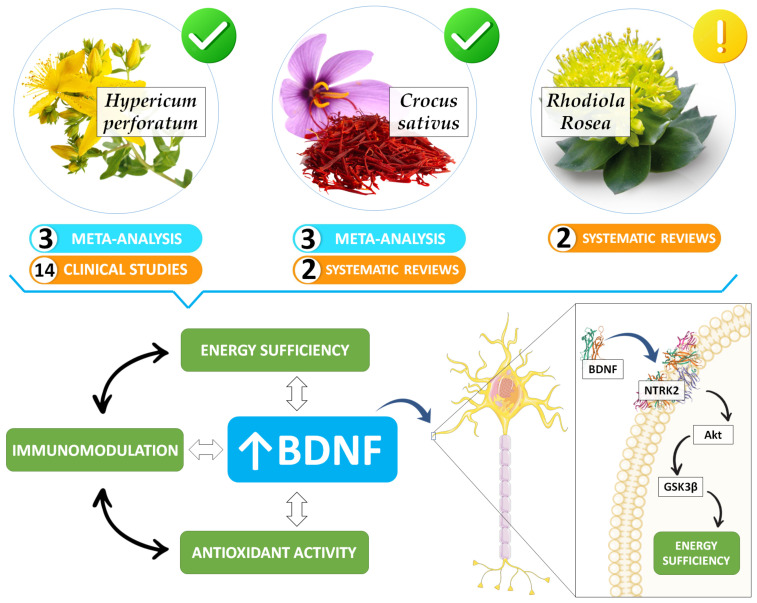
Adaptogens as an antidepressant non-pharmacological strategy. The core of depression is a disorder of metabolism and energy regulation (i.e., allostasis overload) with sensory consequences of that regulation (i.e., altered interoception) which results in a relatively ‘locked-in’ depressed brain. *Hypericum perforatum* and *Crocus sativus* have shown clinically significant benefits on depression-related outcomes; notwithstanding, other herbal adaptogenic extracts require more research before translation into clinical practice (e.g., *Rhodiola rosea*). The protein models shown were taken from the Protein Data Bank archive (https://www.rcsb.org/, accessed on 4 December 2022) and generated at >2.70 Å with X-ray. For brain-derived neurotrophic factor (BDNF), check the UniProtKB: P23560 and PDB ID: 1B8M. For BDNF/NT-3 growth factor receptor (NTRK2), also known as tropomyosin receptor kinase B (TrkB), check the UniProtKB: Q16620 and PDB ID: 1HCF. Source: designed by the authors (D.A.B.) using figure templates developed by Servier Medical Art (Les Laboratoires Servier, Suresnes, France), licensed under a Creative Common Attribution 3.0 Generic License. http://smart.servier.com/ (accessed on 1 December 2022).

**Table 1 ijerph-20-05298-t001:** Synthesis of the selected articles for the integrative review.

Reference	Population *n* (F:M)	Type of Study/ Country	Methodology				Analyzed Outcomes	Main Findings
			Adaptogen	Sample (*n*)	Length	Dosage		
Canning et al. 2010 [48]	32 (32:0) Women diagnosed with mild premenstrual syndrome (18–45 years)	RCT-DB/UK	*Hypericum perforatum*	HP (17) Placebo (15)	10 regular menstrual cycles (25–35 days each)	900 mg∙day^−1^ (Li160: 80% methanolic dry extract, with 0.18% hypericin and 3.38% hyperforin)	BDI, BIS-11, BPAQ, STAIS, hormones and inflammatory cytokines	HP improved physical and behavioral premenstrual symptoms in women with mild premenstrual syndrome. No changes were reported in plasma hormone and cytokines.
Kasper et al. 2010 [49] †	1661 males and females who suffered from a single or recurrent acute episode of mild, moderate or severe major depression	RCT-DB/Germany, France & Sweden	*Hypericum perforatum*	HP_600 (123) HP_900 (945) HP_1200 (127) HP_1800 (69) Paroxetine (126) Placebo (271)	6 weeks	600–1800 mg∙day^−1^ (WS5570: 80% methanolic dry extract, with 0.1–0.3% hypericin and 3–6% hyperforin)	DSM-IV, HAMD	Besides antidepressant efficacy, substantially lower adverse events were evident after HP administration compared to placebo, paroxetine and other SSRI. The ‘typical SSRI side effects’ are less frequent during HP administration.
Mannel et al. 2010 [50]	189 (157:32) Patients diagnosed with atypical depression (18–70 years)	RCT-DB/Germany	*Hypericum perforatum*	HP (91) Placebo (98)	8 weeks	600 mg∙day^−1^ (Li160: 80% methanolic dry extract, with 0.18% hypericin and 3.38% hyperforin)	17-HAMD, HAMD, CGI-S, CGI-I, CGI-E, PHQ-9	Results can be interpreted as a strong support for the efficacy of HP in atypical depression. This beneficial effect was mainly in patients with moderate severity, whereas for patients with a mild severity of major depression no clear benefit could be observed.
Melzer et al. 2010 [51]	1778 (1411:367) patients diagnosed with depressive disorders (18–97 years)	RCT-OL/Germany	*Hypericum perforatum*	HP_270 (452) HP_425 (1319)	12 weeks	600–1000 mg∙day^−1^ (Helarium^®^: ethanolic dry extract with 0.1–0.3% hypericin, 6% hyperforin, and flavonoid/rutoside 6%)	ICD-10, CGI, VAS (subjective well-being)	HP was well tolerated and had no side effects. Despite the study limitations, HP was quite an effective antidepressant for mild to moderate depression.
Dwyer et al. 2011 [52]	Databases: Medline, Cinahl, AMED, ALT Health Watch, Psych Articles, Psych Info, and Cochrane	Systematic review/Australia	Herbal medicines—other than *Hypericum*	Nine RCTs fulfilled inclusion criteria	6–8 weeks	Quality of selected studies was assessed using the Jadad scale.	HAMD	*Crocus sativa* [Saffron] showed consistently positive effects. *Lavendula angustifolia*, *Echium amoenum* and *Rhodiola rosea* deserve future research.
Sacher et al. 2011 [53]	23 (13:10) patients with major depression and healthy controls (19–48 years)	CT/Canada	*Hypericum perforatum*	HP (7) Moclobemide (6) Control (10)	6 weeks	600 mg twice per day	[^11^C]-harmine PET, SCID-P, HAMD,	HP had a negligible effect on MAO-A binding in vivo and, therefore, should not be classified as an MAO-A inhibitor.
Rapaport et al. 2011 [54]	59 female and male patients diagnosed with minor depression (21–82 years)	RCT-DB/USA	*Hypericum perforatum*	HP (22) Citalopram (18) Placebo (19)	12-week	810 mg∙day^−1^ (Cederroth International)	HAMD, GAF, IDS-SR, IDS-CR, Q-LES-Q, CGI, MOS, PRISE	HP has no significant impact on primary and secondary outcomes of minor depressive disorder. However, the effects on physical function were in the HP group when compared to citalopram and placebo.
Singer et al. 2011 [55] †	154 (110:44) outpatients diagnosed with mild to moderate depression	RCT-DB/Germany	*Hypericum perforatum*	HP (54) Citalopram (54) Placebo (46)	6 weeks	900 mg∙day^−1^ (STW 3-VI: 80% ethanolic extract)	HAMD—relapse and recurrence (%)	HP might lower the risk for patients to encounter a relapse and/or recurrence in the time following acute phase treatment of mild to moderate depression, when compared to citalopram.
Sarris et al. 2012 [56]	124 (77:43) DSM-IV diagnosed patients with major depressive disorder	RCT-DB/USA	*Hypericum perforatum*	HP (35) Sertraline (49) Placebo (40)	26 weeks	900–1500 mg∙day^−1^ (Li160: 80% methanolic dry extract, with 0.12–0.28% hypericin)	17-HAMD, BDI, GAF, CGI-S, CGI-I	Data revealed an equivocal outcome between treatments at week 26, both HP and sertraline were still therapeutically effective, with a pronounced “placebo-effect” impeding a significant result at week 26.
Hausenblas et al. 2013 [57]	Databases: Allied and Complementary Medicine database, Cumulative Index to Nursing and Allied Health Literature, The Cochrane Library, EMBASE, MEDLINE, PubMed, and Web of Science	Systematic review with meta-analysis/Australia	*Crocus sativus* L.	Five RCTs fulfilled inclusion criteria	6–8 weeks	30 mg∙day^−1^. Quality of selected studies was assessed using the Jadad scale.	DSMI-IV, HAMD	Reviewed articles indicate that saffron supplementation can improve symptoms of depression in adults with major depressive disorder. However, authors stated the need for larger clinical trials following CONSORT guidelines and outside Iran.
Grobler et al. 2014 [58] †	340 (224:116) outpatients diagnosed with major depressive disorder	RCT-DB/USA	*Hypericum perforatum*	HP (113) Sertraline (111) Placebo (116)	8 weeks (optional follow-up phase to 26 weeks)	900–1500 mg∙day^−1^ (Li160: 80% methanolic dry extract, with 0.12–0.28% hypericin)	HAMD, GAF, CGI-S, CGI-I, BDI + Blood chemistry and hematology	No significant difference was established between HP and placebo, but there was a significant difference between the sertraline and placebo. This re-analysis of the original data changes the original conclusion of the trial.
Ross et al. 2014 [59]	89 both male and female patients diagnosed with mild to moderate depression (18–70 years)	RCT-DB/USA	*Rhodiola rosea* L.	RR_340 (31) RR_580 (29) Placebo (29)	6 weeks	340–680 mg∙day^−1^ (SHR-5 powered root 70% ethanolic extract [4:1])	BDI, HAMD	340 mg∙day^−1^ of RR over a 6-week period significantly reduced overall symptoms of depression. Similar results were found in the group that consumed 680 mg of RR but, in addition, there was a significant improvement in self-esteem parameters.
Lee & Ji 2014 [60]	93 postmenopausal women (50–73 years)	RCT-DB/Korea	Fermented red ginseng	Ginseng (49) Placebo (44)	2 weeks	2.1 g∙day^−1^ (lyophilized FRG powder)	BDI + Blood chemistry	A significant difference was found on cognitive depression between the RR and placebo groups, possibly via the energy factor.
Lopestri & Drummond 2014 [61]	Databases: PubMed/Medline, Google Scholar, PsycINFO and The Cochrane Library	Systematic review/Australia	*Crocus sativus* L.	Six RCTs fulfilled inclusion criteria	6–8 weeks	30 mg∙day^−1^ Quality of selected studies was assessed using the Jadad scale.	DSM-IV and 17-HAMD	Saffron is as a promising natural option for the treatment of mild-to-moderate depression with initial clinical research supporting its efficacy, at least in the short term.
Chen et al. 2015 [62] †	277 (183:94) outpatients diagnosed with major depressive disorder	RCT-DB/USA	*Hypericum perforatum*	HP (88) Sertraline (90) Placebo (99)	8 weeks	900–1200 mg∙day^−1^ (Li160: 80% methanolic dry extract, with 0.12–0.28% hypericin)	17-HAMD, GAF, CGI-S, CGI-I, BDI	In the comparative efficacy of HP for treatment of major depressive disorder, clinicians’ guesses regarding subjects’ treatment assignments were significantly associated with improvement in 17-HAMD scores and remission rates, regardless of treatment received.
Cropley et al. 2015 [63]	80 (48:32) mildly anxious participants (score above 30 on the STAI)	RCT-OL/UK	*Rhodiola* *rosea* L.	RR (39) Control (41)	14 days	400 mg∙day^−1^ (Vitano^®^: proprietary dry extract from Rhodiola rosea roots (1.5–5:1) rosalin WS^®^ 1375)	STAI, POMS, Perceived Stress Scale	After 14 days of intervention, RR was well tolerated and demonstrated to significantly lower ratings of self-reported anger, confusion, and depression relative to the controls and showed significant improvements in total mood, over the course of the study.
Stojanovska et al. 2015 [64]	29 (29:0) Chinese postmenopausal women	RCT-DB/China	*Lepidium* *meyenii*	Maca (14) Placebo (15)	6 weeks	3.3 g∙day^−1^ (Maca Power, Murwillumbah, Australia: 1386 mg net Maca)	GCS, SF-36 v2, WHQ, UQoL + Blood chemistry	Neither estrogenic nor immune effects were found; however, Maca was shown to be effective in reducing blood pressure and depression in postmenopausal women.
Mao et al. 2015 [65]	57 (26:31) outpatients diagnosed with major depressive disorder	RCT-DB/USA	*Rhodiola* *rosea* L.	RR (20) Sertraline (19) Placebo (18)	12 weeks	340–680 mg∙day^−1^ (SHR-5 powered extract with 3.07% rosavin and 1.95% rhodioloside)	17-HAMD, CGI-C, BDI	Findings suggest that RR might possess modest antidepressant effects in some patients with mild-to-moderate major depressive disorder. In addition, RR may be better tolerated than conventional anti-depressants.
Jeong et al. 2015 [66]	35 (35:0) outpatients diagnosed with major depressive disorder	CT-OL/Korea	Korean Red Ginseng	Ginseng (35)	8 weeks	2–3 g∙day^−1^ (lyophilized powder extract)	DRSS, DSSS, MADRS, CGI-S + Blood chemistry	Korean Red Ginseng treatment for patients with residual depression symptoms seems to be effective and safe.
Amsterdam et al. 2016 [67]	Databases: BIOSIS, CAplus, TOXCENTER, EMBASE, NAPRALET, PubMed/Medline, and Russian State Library in Moscow	Systematic review/USA & Sweden	*Rhodiola* *rosea* L.	Two RCTs and seven CT-OL were reviewed	2–12 weeks	340–680 mg∙day^−1^ Quality of selected studies was assessed using the Jadad scale.	DSM-IV and 17-HAMD	The reviewed literature suggests a possible antidepressant action for RR extract in adult humans. In contrast to most conventional antidepressants, RR extract appears to be well-tolerated in short-term studies with a favorable safety profile.
Seifritz et al. 2016 [68] †	64 (39:25) outpatients diagnosed with moderate depression	RCT-DB/Germany	*Hypericum perforatum*	HP (31) Paroxetine (33)	6 weeks	900 mg∙day^−1^ (WS5570: 80% methanolic dry extract, with 0.1–0.3% hypericin and 3–6% hyperforin)	HAMD— response and remission	A daily dose of 900 mg of HP extract was significantly superior to the SSRI (paroxetine 20 mg) with respect to the reduction in the HAMD score during the treatment period.
Apaydin et al. 2016 [69]	Databases: PubMed/Medline CINAHL, Embase, PsycINFO, AMED, CENTRAL, Web of Science, MANTIS, and ICTRP	Systematic review with meta-analysis/USA	*Hypericum perforatum*	Thirty-five RCTs fulfilled inclusion criteria	4–32 weeks	100–1200 mg∙day^−1^ Cochrane Risk of Bias tool, the USPSTF criteria, and the GRADE approach	HAMD—remission and relapse	The available evidence suggests that HP extracts are effective in treating patients with mild and moderate major depressive disorder compared to placebo and to antidepressants. Fewer side effects were reported compared to antidepressants.
Kell et al. 2017 [70]	121 (75:46) patients diagnosed with mild-to- moderate depression (18–77 years)	RCT-DB/Australia	*Crocus sativus* L.	Saffron_22 (42) Saffron_28 (41) Placebo (38)	4 weeks	22–28 mg∙day^−1^ (affron^®^: capsule with 11–14 mg of standardized saffron extract containing >3.5% lepticrosalides^®^ from Pharmactive Biotech Products)	DASS-21, POMS, PANAS, PSQI	A significant decrease in negative mood and symptoms related to stress and anxiety were found after 28-mg/day saffron supplementation. Similarly, sleep quality showed a slight improvement only at 28 mg/day dose.
Ng et al. 2017 [71]	Databases: PubMed/Medline Ovid, CCDANTR, CFCM, CNKI, and WanFang	Systematic review with meta-analysis/Singapore	*Hypericum perforatum*	Twenty-seven RCTs and CTs fulfilled inclusion criteria	4–12 weeks	20–1350 mg∙day^−1^ Cochrane Risk of Bias tool and the Jadad scale	HAMD, CGI, MADRS, BDI	The findings further strengthen the support for HP clinical efficacy in alleviating depressive symptoms. In patients with mild-to-moderate depression, HP demonstrated comparable response and remission rates, and a significantly lower discontinuation/dropout rate compared to standard SSRIs.
Tóth et al. 2018 [72]	Databases: PubMed/Medline, Embase, Cochrane, and Web of Science	Systematic review with meta-analysis/Hungary	*Crocus* *sativus* L.	Eleven RCTs fulfilled inclusion criteria	4–32 weeks	20–100 mg∙day^−1^ Cochrane Risk of Bias tool	HAMD, BDI	As stated by the authors, results clearly suggest that saffron reduces the severity of depression, but the optimum dose and duration of treatment are still unclear.
Murck et al. 2018 [73] †	247 (183:64) patients diagnosed with mild to moderate depression	RCT-DB/UK	*Hypericum perforatum*	HP (123) Placebo (124)	6 weeks	900 mg∙day^−1^ (Li160: 80% methanolic dry extract, with 0.12–0.28% hypericin)	17-HAMD, MADRS + electrolyte-related analyses	No difference was found between HP and placebo. Analysis of the pooled dataset showed that low Na^+^/K^+^ ratio and high [K^+^] was associated with a worse depression-related outcome after 6 weeks of intervention.
Di Pierro et al. 2018 [74]	60 (30:30) patients diagnosed with moderate depression	CT-OL/Italy	*Hypericum perforatum*	Mono_HP (30) Multi_HP (30)	48 weeks	600 mg∙day^−1^ (Nervaxon^®^ [mono-fractioned] and IperiPlex^®^ [multi-fractioned]: both extracts with 0.3% hypericin)	ZDS + physiological and hormonal analyses + blood chemistry	A multi-fractioned HP extract has better clinical outcomes in subjects with depression than mono-fractioned extract without increases in toxicity or reduced tolerability. No side effects were reported.
Eatemadnia et al. 2019 [75]	70 (70:0) Iranian postmenopausal women	RCT-DB/Iran	*Hypericum perforatum*	HP (35) Placebo (35)	8 weeks	0.990 mg∙day^−1^ (Gole Darou Company, Isfahan, Iran).	21-HAMD & modified Kupperman index	HP might significantly reduce the depression score in post-menopausal women after eight weeks of treatment.
Ganon et al. 2019 [76]	59 female and male outpatients diagnosed with schizophrenia or schizoaffective disorder	RCT-DB secondary analysis/USA	*Withania somnifera*	WSE (28) Placebo (31)	12 weeks	500–1000 mg∙day^−1^ (Sensoril^®^: standardized extract with ≥8% withanolides and ≤2% withaferin A)	PANASS	Findings suggest that WSE may hold promise in the treatment of depression and anxiety symptoms in schizophrenia. Although gastrointestinal distress and somnolence were more commonly reported in the WSE group, there were no differences against placebo.
Lopestri et al. 2019 [77]	60 (23:37) mildly anxious participants (score between 6–17 on the HAM-A)	RCT-DB/India	*Withania somnifera*	WSE (30) Placebo (30)	8 weeks	240 mg∙day^−1^ (Shoden^®^: 70% ethanolic extract with 84 mg of withanolide glycosides)	HAMA, DASS-21 + Hormonal analysis + Blood chemistry	WSE has positive anxiolytic effects after 60 days at a dose of 240 mg daily. WSE was associated with reduced cortisol and DHEA-S. No reported significant adverse effects.
Gao et al. 2020 [78]	98 (53:45) outpatients diagnosed with mild-to-moderate major depressive disorder	RCT-DB/China	*Rhodiola rosea* L.	RR_300 (33) RR_600 (33) Sertraline (32)	12 weeks	300–600 mg∙day^−1^ (Tibet GaoYuanan Biotechnology Co., Ltd., Tibet, China)	HAMD, BDI, CGI-C + Hormonal analysis + Blood chemistry	Findings suggest that RR might have antidepressant effects in patients with mild-to-moderate major depressive disorder. RR was better tolerated than sertraline.
Dai et al. 2020 [79]	Databases: PubMed/Medline, Embase, and ScienceDirect	Systematic review with meta-analysis/China	*Crocus sativus* L.	Twelve RCTs fulfilled inclusion criteria	1–12 weeks	20–50 mg∙day^−1^ Cochrane Risk of Bias tool	HAMD, BDI	Overall results showed that saffron exerted superior effect compared with placebo and that it might be as effective as antidepressants (fluoxetine or citalopram) in alleviating depressive symptoms. Saffron seemed to be safe and well tolerated.
Lee et al. 2020 [80]	36 (27:9) difficult-to-treat patients with major depressive disorder	CT-OL/Korea	Korean Red Ginseng	Ginseng (36) 28 patients had post-baseline visit data	6 weeks	2 g∙day^−1^ (red ginseng extract with ginsenosides)	MADRS, PHQ-15, PSS, CGI-I	This study indicates the putative effectiveness and tolerability of red Ginseng for treating patients diagnosed with difficult-to-treat major depressive disorder.
Konstantinos & Heun 2020 [81]	Databases: PubMed/Medline	Systematic review/Greece & Germany	*Rhodiola rosea* L.	Five RCTs fulfilled inclusion criteria	2–6 weeks	100–1000 mg∙day^−1^ Cochrane Risk of Bias tool	HAMD, BDI, MADRS, STAI, POMS, SAM, SF-36, CGI-I	RR supplementation might alleviate symptoms of mild to moderate depression and mild anxiety, while it may also enhance mood.
Zhou et al. 2021 [82]	48 female and male patients diagnosed with moderate depression	RCT-DB/China	*Cordyceps militaris*	Cordyceps (23) Placebo (25)	6 weeks	2 g∙day^−1^ (red ginseng extract with ginsenosides)	HAMD, AIS	A significant improvement in HAMD-17 scores was found in both groups; however, there was no significant difference between the effects of treatment with duloxetine + cordyceps or duloxetine + placebo on the depression-related outcomes.
Tammadon et al. 2021 [83]	39 (19:20) hemodialysis patients	RCT-DB/Iran	*Valeriana officinalis*	Valerian (20) Placebo (19)	4 weeks	530 mg∙day^−1^ (Dried root extract)	PSQI, BDI, STAI	Valerian significantly improved sleep quality, the symptoms of state anxiety, and depression in hemodialysis patients.
Paul et al. 2021 [84]	Comprehensive review focused on phytochemistry, clinical and toxicological aspects	Literature review/India	*Withania somnifera*	No information of inclusion criteria	1–42 weeks	100–3000 mg (root extract)	Subjective scales + Inflammatory markers + Antioxidant	Based on pre-clinical and clinical research, WSE has been found to be especially active against many neurological and psychological conditions including depression.
Lopresti & Smith 2021 [85]	Databases: PubMed/Medline CINAHL, Cochrane Library, Scopus, Web of Science	Systematic review/Australia	*Withania somnifera*	Forty-one RCTs fulfilled inclusion criteria	4–32 weeks	100–1200 mg∙day^−1^ Cochrane Risk of Bias tool and the Newcastle-Ottawa scale	PSS, HAMA, DASS-21, POMS, CGI + Hormonal analyses + Blood chemistry	Overall, the strongest evidence for therapeutic efficacy of WSE is the alleviation of stress and anxiety symptoms. However, few studies have evaluated the impact on depression-related symptoms.
Speers et al. 2021 [86]	Databases: PubMed/Medline, Scopus, Google Scholar	Literature review/USA	*Withania somnifera*	No information of inclusion criteria	4–32 weeks	120–1000 mg∙day^−1^ (leaf and root extracts)	HAMA, PSS, PANSS, PSQI, QoL, DASS	Additional rigorous studies using standardized WSE products and doses in populations diagnosed with depression rather than healthy participants are needed.
Kenda et al. 2022 [87]	Review focused on plants and derived products used for anxiety, depression or stress	Literature review/Slovenia	Herbal medicines	No information of inclusion criteria	NA	Saffron: 30–200 mg∙day^−1^ HP: 500–1800 mg∙day^−1^	NA	*Crocus sativus* L. and *Hypericum perforatum* are effective in ameliorating depression and its use is beneficial in patients with mild to moderate depression.
Zhao et al. 2022 [88]	Databases: PubMed/Medline, CINAHL, Scopus, Web of Science	Systematic review with meta-analysis/China	*Hypericum perforatum*	Fourteen RCTs fulfilled inclusion criteria	6–26 weeks	300–1800 mg∙day^−1^ Cochrane Risk of Bias tool	HAMD	HP is a cheap, readily available and effective treatment strategy for mild-to-moderate depression. Based on the results of this meta-analysis, the use of HP for depression is strongly recommended.

Data are expressed as means (SD). Participants represent those that were included in the final analysis. [^11^-C] PET: [^11^C]-harmine positron emission tomography; IDS-CR: 30-item Inventory of Depressive Symptomatology-Clinician Rated scale; AIS: Athens Insomnia Scale; BDI: Beck Depression Inventory; BIS-11: Barratt Impulsiveness Scale version 11; BPAQ: Aggression Questionnaire; CGI: Clinical Global Impression scale; CGI-C: Clinical Global Impression Change; CGI-E: Clinical Global Impression-Efficacy; CGI-I: Clinical Global Impression-Improvement; CGI-S: Clinical Global Impression-Severity of Illness; CT: clinical trial; DASS-21: 21-Items Depression, Anxiety and Stress Scale; DRSS: Depression Residual Symptom Scale; DSSS: Somatic Symptoms and Depression Scale; GAF; Global Assessment of functioning scale; GCS: Greene Climacteric Scale; HAMA: Hamilton Anxiety Rating Scale; HAMD: Hamilton Scale of Depression; HP: *Hypericum Perforatum*; IDS-SR: 30-item Inventory of Depressive Symptomatology-Self Report scale; MADRS: Montgomery–Åsberg Depression Rating Scale; MOA-A: monoamine oxidase A; MOS: Medical Outcomes Study 36-item short-form; NA: not available; OL: open labelled; PANAS: Positive and Negative Affect Schedule; PHQ: Patient Health Questionnaire; POMS: profile of mood states; PRISE: Patient-Rated Inventory of Side Effects; PSQI: Pittsburgh Sleep Quality Index; PSS: Patient Satisfaction Score; Q-LES-Q: Quality of Life, Enjoyment and Satisfaction Questionnaire; RR: *Rhodiola rosea*; RCT-DB: double-blinded randomized clinical trial; SCID-P: Structured Clinical Interview for DSM-IV, Patient Edition; SSRI: Selective Serotonin Reuptake Inhibitors; STAI: State-Trait Anxiety Inventor; VAS: visual analogue scale; WHQ: Women’s Health Questionnaire; UQoL: Utian Quality of Life; WSE: *Withania somnifera* (ashwagandha) extract; ZDS: Zung Depression Self-Rafting Scale. † Re-analyzed data.

## Data Availability

Not applicable.

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
