# Peer review of "Adaptogens on Depression-Related Outcomes: A Systematic Integrative Review and Rationale of Synergism with Physical Activity"

_ijerph, 2023, doi:10.3390/ijerph20075298_

Round 1

Reviewer 1 Report

I believe the topic is timely and important and I thank the authors for their interest and time studying this topic. I thought the authors did a good job laying out the methods and took extra care to perform a systematic review. I think there could be more done on the meta-analysis part. There is good information presented but the discussion could be more direct and reflective of the findings.

Most importantly, the introduction needs major revision. The writing seems sterile at the beginning, lacks flow, and is hard to follow. It seems to be pieced together from other research writings which dilutes any continuity for the reader. I would recommend a single editor to review and rewrite the paper (especially the Intro) so one overall writing style is clear and apparent

LIne 52 - did you mean "personal life" and not "personal level"?

Line 396-398 - you finally define "allostatic load" but I think you should put this definition early in the paper as well (i.e., line 61). You explain what it is on page 2 but a definition to start with would be helpful.

Also, consider defining/explaining other research terms more clearly in the introduction (i.e., adaptogen [line 92], the three ingredients contributing to depression [lines 82-84], etc.).

Author Response

Dear reviewer, 

Thanks for your report.
Please see attached the file with point-by-point responses. 

Sincerely,
The authors

Reviewer 2 Report

In this systematic integrative review, the authors aimed to the effect of adaptogen supplementation on depression-related symptoms and signs in adults, and to discuss the possible combination with exercise to help plan and commission future clinical research. The overall quality of the manuscript is good, however, it should be revised to improve its quality.

·         The introduction should elaborate on the pathogenesis of depression. 

·         What is the rationale and novelty of the present study?

·     What benefits of adaptogens were expected compared to other commercially available medicines used for depression?

·         The discussion should summarize the findings of the study and connect the different results.

·         Several grammatical issues need to be fixed.

·         Specify your abbreviations along with the whole manuscript.

·         What are your significant values?

Author Response

(The authors gave the same response as above.)

Reviewer 3 Report

To recheck the bibliography for self-citations and a decrease in their number.

Author Response

Dear reviewer, 

Thanks for your comment. 
This has been amended as requested.

Sincerely,
The authors